# Goal-Directed Fluid Therapy Enhances Gastrointestinal Recovery after Laparoscopic Surgery: A Systematic Review and Meta-Analysis

**DOI:** 10.3390/jpm12050734

**Published:** 2022-04-30

**Authors:** Marcell Virág, Máté Rottler, Noémi Gede, Klementina Ocskay, Tamás Leiner, Máté Tuba, Szabolcs Ábrahám, Nelli Farkas, Péter Hegyi, Zsolt Molnár

**Affiliations:** 1Szentágothai Research Centre, Institute for Translational Medicine, Medical School, University of Pécs, 7624 Pécs, Hungary; viragmarcell@yahoo.com (M.V.); mate.rottler@gmail.com (M.R.); gede.noemi@gmail.com (N.G.); ocskay.klementina@gmail.com (K.O.); tamasleiner@gmail.com (T.L.); mate.tuba@gmail.com (M.T.); szabolcs.abraham@gmail.com (S.Á.); farkas.nelli@gmail.com (N.F.); hegyi2009@gmail.com (P.H.); 2Department of Anesthesiology and Intensive Therapy, Szent György University Teaching Hospital of Fejér County, 8000 Székesfehérvár, Hungary; 3Doctoral School of Clinical Medicine, University of Szeged, 6720 Szeged, Hungary; 4Centre for Translational Medicine, Semmelweis University, 1085 Budapest, Hungary; 5Anaesthetic Department, Hinchingbrooke Hospital, North West Anglia NHS Foundation Trust, Huntingdon PE29 6NT, UK; 6Division for Pancreatic Disorders, Heart and Vascular Center, Semmelweis University, 1122 Budapest, Hungary; 7Department of Anaesthesiology and Intensive Therapy, Poznan University of Medical Sciences, 61-701 Poznan, Poland; 8Department of Anaesthesiology and Intensive Therapy, Semmelweis University, 1082 Budapest, Hungary

**Keywords:** enhanced recovery after surgery, goal-directed fluid therapy, intraoperative fluid management, haemodynamic monitoring, laparoscopic abdominal surgery, perioperative care

## Abstract

(1) Background: Whether goal-directed fluid therapy (GDFT) provides any outcome benefit as compared to non-goal-directed fluid therapy (N-GDFT) in elective abdominal laparoscopic surgery has not been determined yet. (2) Methods: A systematic literature search was conducted in MEDLINE, Embase, CENTRAL, Web of Science, and Scopus. The main outcomes were length of hospital stay (LOHS), time to first flatus and stool, intraoperative fluid and vasopressor requirements, serum lactate levels, and urinary output. Pooled risks ratios (RRs) with 95% confidence intervals (CI) were calculated for dichotomous outcomes and weighted mean difference (WMD) with 95% CI for continuous outcomes. (3) Results: Eleven studies were included in the quantitative, and fifteen in the qualitative synthesis. LOHS (WMD: −1.18 days, 95% CI: −1.84 to −0.53) and time to first stool (WMD: −9.8 h; CI −12.7 to −7.0) were significantly shorter in the GDFT group. GDFT resulted in significantly less intraoperative fluid administration (WMD: −441 mL, 95% CI: −790 to −92) and lower lactate levels at the end of the operation: WMD: −0.25 mmol L^−1^; 95% CI: −0.36 to −0.14. (4) Conclusions: GDFT resulted in enhanced recovery of the gastrointestinal function and shorter LOHS as compared to N-GDFT.

## 1. Introduction

Laparoscopic surgical techniques have become the first choice over the last decade due to the lower incidence of postoperative surgical complications, faster recovery, and less postoperative pain compared to the traditional open techniques [1]. Although the surgical trauma is substantially less with laparoscopic than with open surgery [2,3], the increased intraabdominal pressure caused by the insufflation of the peritoneum can lead to hemodynamic instability, resulting in unfavourable neuroendocrine responses and outcomes [4,5]. Furthermore, laparoscopic surgery may lengthen procedural time compared to laparotomy, which can also pose a special challenge during the anaesthetic management [6].

It is well known that inappropriate intraoperative fluid administration as part of hemodynamic management increases the rates of postoperative complications, could delay the recovery of gastrointestinal function, and therefore may lead to prolonged length of hospital stay [7]. The Early Recovery After Surgery (ERAS) Society highlights the importance of fluid management in its guidelines. Inadequate and/or uncontrolled fluid administration can lead to unnecessary fluid restriction or fluid overload [8,9,10]. The elevated intraabdominal pressure and Trendelenburg and reverse Trendelenburg positions during laparoscopic surgery may reduce renal and splanchnic blood flow [11,12]. This phenomenon is potentially escalated by inadequate intraoperative fluid administration, especially in vulnerable patients suffering from chronic cardiovascular diseases and obesity [13,14].

Defining the appropriate amount of fluid for an individual patient is not easy. One of the potential alternatives is goal-directed fluid therapy (GDFT) [15]. The beneficial effects of GDFT on postoperative complications in high-risk surgical patients have been shown in previous meta-analyses [16,17]. However, a recent randomised clinical trial was unable to demonstrate the clinical benefit of GDFT in elective colectomy patients managed according to the ERAS guideline [18]. Therefore, which patients would benefit from GDFT within the context of the ERAS concept remains unclear [19].

The recent meta-analyses assessing GDFT and N-GDFT [20,21,22,23,24,25,26,27,28] and the guidelines for intraoperative fluid therapy of ERAS and the American Society for Enhanced Recovery do not contain clear recommendations for fluid therapy in laparoscopic surgery [29,30,31]. Therefore, we decided to conduct a systematic review and meta-analysis to assess the effects of GDFT on several postoperative outcomes in patients undergoing elective laparoscopic abdominal surgery.

## 2. Materials and Methods

### 2.1. Registration and Protocol

Our systematic review and meta-analysis were reported according to the Preferred Reporting Items for Systematic Reviews and Meta-Analyses (PRISMA) [32]. The study protocol was registered with the International Prospective Register of Systematic Reviews (PROSPERO) in January 2021 (CRD42021230286). After gathering the statistical results for urinary output and operation time, we considered it feasible to standardise the intraoperative urinary output to the length of the operation, which gave more adequate information about the diuresis per hour of the patients, and thus we decided to deviate from the previously registered protocol in this particular case. There were no other deviations.

### 2.2. Eligibility Criteria

Goal-directed fluid therapy (GDFT) versus non-goal-directed fluid therapy (N-GDFT) was compared in adults undergoing abdominal laparoscopic surgery. Only randomised controlled trials (RCTs) were eligible for inclusion. GDFT was defined as the protocolised administration of fluids and vasoactive and inotropic agents on the basis of haemodynamic assessment, targeted to reach the therapeutic goals [33]. A list of accepted haemodynamic measurement devices is shown in Appendix A. Central venous pressure (CVP)-guided fluid therapy was not considered as GDFT [34]. All laparoscopic abdominal, urological, and gynaecological surgical interventions were included in the analysis, except for laparoscopic cholecystectomy, where we did not consider the application of advanced haemodynamic monitoring feasible due to the short operation times. RCTs reporting data for laparoscopic subgroups separately were also accepted.

### 2.3. Data Items

The following outcomes were evaluated: length of hospital stay (days) defined as the time elapsed between the admission and discharge of the patients, 30-day readmission rate (percentage), reoperation rate (percentage), overall complications (number of patients with at least one undesired event defined by the individual study) within 30 days, appearance of the time to first flatus and first stool after the intervention (hours), intraoperatively administered fluids (mL), number of patients receiving any vasoactive agents during the intraoperative period, urinary output standardised to the length of surgery (mL h^−1^), and serum lactate level at the end of the operation (mmol L^−1^).

### 2.4. Search Strategy and Information Sources

A systematic search was conducted in MedLine via PUBMED^®^, Embase^®^, Cochrane Central Register of Controlled Trials (CENTRAL), Web of Science, and SCOPUS^®^ without any language restrictions and date filters. The last search took place on the 16th of December 2021. Our search key is included in Appendix A.

### 2.5. Selection Process

Duplicates were removed by using a reference management software (EndNote X9, Clarivate Analytics). Title and abstract, and finally the full-text selection were conducted independently by two of the authors (M.V. and M.R.) according to the predefined eligibility criteria. To measure inter-rater reliability, Cohen’s kappa was calculated at the end of each selection step, and the calculated values were considered between 0.41 and 0.60 as moderate, 0.61 and 0.80 as substantial, and 0.81 and 1 as an almost perfect agreement [35]. In the case of a discrepancy, conflicts were resolved by a third review author (K.O.). Reference lists of eligible studies to the qualitative synthesis were also assessed manually to identify any additional records.

### 2.6. Data Collection Process

The following data were collected by M.V. and T.L. independently into standardised electronic spreadsheets in Microsoft Excel 2019^®^ (Microsoft, Redmond, WA, USA), including characteristics of studies (year of publication, number of centres and country), demographic data of patients (i.e., age, Physical Status Classification System of the American Society of Anesthesiology), type and duration of surgery, characteristics of the induction and the maintenance of anaesthesia, aspects of perioperative treatment including protocol of goal-directed fluid regimen and the applied haemodynamic devices in the intervention group, protocol of fluid administration in the control group, protocol of pre- and postoperative fluid therapy, postoperative overall complications with predefined criteria of the studies, length of hospital stay, quantity and type of fluids administered intraoperatively, intraoperative urinary output, lactate levels at the end of the operation, and time to first flatus and stool in the postoperative period.

### 2.7. Synthesis of Results and Effect Measures

Forest plots were used to display the results of the meta-analysis. Pooled risk ratios (RRs) with 95% confidence intervals (CI) were calculated for dichotomous outcomes and weighted mean difference (WMD) with 95% CI for continuous outcomes. In the case of urinary output, standardised mean difference (SMD) was calculated for the average operation length. Data were converted from median and first and third quartile to mean and standard deviation, if data were reported in the former, according to the method of Wan (2014) [36]. Sensitivity analyses were also carried out, omitting one study and calculating the summary of RR, WMD, or SMD with 95% CI to investigate the influence of a single study on the final estimation.

A random-effect model was applied in all analyses with the estimation of DerSimonian and Laird [37]. Statistical heterogeneity was analysed using the I2 and χ2 tests to gain probability values; *p* < 0.10 was defined to indicate significant heterogeneity. The I2 test represents the percentage of total variation across studies because of heterogeneity. I2 values of 25–50%, 50–75%, and 75–100% corresponded to low, moderate, and high heterogeneity, respectively, on the basis of the Cochrane’s handbook [38]. All data management and statistical analyses were performed with Stata 16 SE (Stata Corp, College Station, TX, USA).

### 2.8. Study Risk of Bias Assessment and Reporting Bias Assessment

To identify the risk of bias of the included studies, two review authors (M.V. and M.R.) used RoB 2, a revised Cochrane Collaboration’s risk of bias tool for randomised trials [39]. The included studies were evaluated according to all five domains for each outcome (randomisation process, deviations from intended interventions, missing outcome data, measurement of the outcome, selections of the reported result) and finally, the overall risk of bias was classified as low, some concerns, or high. Discrepancies were resolved by a third review author (K.O.). The presence of publication bias was assessed by visual inspection of Funnel plots for lack of asymmetry [40]. The Egger’s test was not performed due to the low number of studies.

### 2.9. Certainty Assessment

Quality of evidence (QoE) was evaluated by M.V. and supervised by Z.M. with the help of the GRADE profiler (GRADEpro) according to the GRADE approach recommended by the Cochrane Collaboration [41,42,43]. The following domains were appraised: risk of bias; indirectness of evidence; serious inconsistency; imprecision of effect estimates; and other considerations such as publication bias, large effect, and plausible confounding.

## 3. Results

### 3.1. Study Selection

The selection process is detailed in Figure 1. Our search resulted in 5485 records from five databases (PUBMED^®^, Embase^®^, CENTRAL, Web of Science, and SCOPUS^®^). Finally, 15 RCTs were included in the qualitative synthesis [44,45,46,47,48,49,50,51,52,53,54,55,56,57,58], and in one RCT (Cho and colleagues), two types of GDFT protocols were implemented [46]. According to our selection criteria, it was not possible to decide which group should be included in the quantitative synthesis, and therefore we decided to include this study only in the qualitative synthesis. Eleven studies were included in the quantitative synthesis [47,48,49,50,51,52,53,54,56,57,58].

### 3.2. Study Characteristics

Baseline characteristics and demographic data for the included studies are presented in Table 1 and Appendix A. A total of 835 patients from 11 studies were included in the quantitative synthesis. During the intraoperative period, 419 patients received GDFT, and 416 received N-GDFT. Nine out of eleven studies reported data on age; the mean age was 55.6 years in the GDFT group and 54.8 years in the N-GDFT group. The rest of the patients’ baseline characteristics and length of operation are detailed in Appendix A.

### 3.3. Results of Syntheses and Individual Studies

#### 3.3.1. Length of Hospital Stay

Length of hospital stay was significantly shorter in the GDFT group (WMD: −1.18 days, 95% CI: −1.84 to −0.53) according to data from eight RCTs [48,49,50,51,52,54,56,58], but data were considered highly heterogeneous (I2 = 80.1%, *p* < 0.01) (Figure 2). As an implementation of the ERAS protocol could have a substantial effect on hospital stay, we performed subgroup analysis on studies that used ERAS protocols and those that did not. Only three studies implemented ERAS [48,54,56], and no significant difference was found between GDFT and N-GDFT (WMD: −1.18 days, 95% CI: −2.79 to 0.43). In those studies that did not use the ERAS protocol, a significant difference was detected between the two groups (WMD: −1.28 days, 95% CI: −2.12 to −0.44); however, high heterogeneity was detected (I2 = 85.5%, *p* < 0.01). No influential study was identified by the leave-one-out sensitivity analysis (Appendix A). Data for length of hospital stay were presented in one further study (Cho and colleges) that was not included in our meta-analysis [46], for reasons detailed previously. Nevertheless, no significant differences were observed between the two goal-directed groups and the controls (4.40 and 4.40 days in the two GDFT groups versus 4.52 days in the non-goal-directed group, *p* = 0.78).

#### 3.3.2. Readmission and Reoperation Rate

A 30-day readmission to the surgical ward and the emergency department were detailed only by Gomez-Izquierdo et al. [48]. No significant differences were found (8 out of 64, 12.0% versus 6 out of 64, 9.4%, *p* = 0.35; 3 out of 64, 20.0% versus 9 out of 64, 14.0%, *p* = 0.58, respectively). The reoperation rate was reported by both Gomez-Izquierdo et al. and Joosten et al. [48,49]. No significant differences were found between the GDFT and N-GDFT groups (1 out of 19, 5.0% versus 2 out of 20, 10.0%, *p* = 0.58; 1 out of 64, 3.1% and 3 out of 64, 4.7%, *p* = 0.62, respectively).

#### 3.3.3. Overall Complications within 30 Days

Nine studies reported data for overall complications [44,45,48,49,51,54,56,57,58]; however only two fulfilled our criteria [44,48], and hence we were unable to perform a quantitative synthesis. In these two studies, there were no significant differences regarding this outcome (43.8% versus 39.1%, *p* = 0.59; 28.1% versus 26.3%, *p* = 0.86, respectively).

#### 3.3.4. Recovery of Gastrointestinal Function as Indicated by Time to Firs Flatus and Time to First Stool

Five trials evaluated time to first stool [49,51,52,57,58], which was significantly reduced in patients receiving GDFT (WMD: −9.8 h, 95% CI: −12.7 to −7.0; Figure 3A). The leave-one-out sensitivity analysis did not identify any influential study (Appendix A). Five further studies reported time to first flatus with significant difference between the two groups (WMD: −5.63 h, 95% CI: −10.9 to 0.4 h) [48,49,50,56,57], but heterogeneity was high (I2 = 92.0%, *p* < 0.01; Figure 3B). According to the leave-one-out sensitivity analysis, omission of studies published by Tang, Wen, and Li would change the statistical significance (Appendix A).

#### 3.3.5. Intraoperative Clinical Outcomes: Intraoperative Fluid and Vasopressor Requirement, Standardised Intraoperative Urinary Output and Lactate Levels at the End of the Operation

Data for clinical outcomes are shown in Figure 4. According to seven studies reporting data for intraoperative fluid requirement [48,49,51,53,56,57,58], patients undergoing GDFT received significantly less fluid than controls (WMD: −441 mL, 95% CI: −790 to −92 mL), with high heterogeneity (I2 = 96.9%, *p* < 0.01). Leave-one-out sensitivity analysis did not report any influential study (Appendix A). Cho and colleges reported similar data indicating that significantly less fluid (colloid) boluses were administered in the GDFT group versus controls (858 mL versus 1639 mL; *p* < 0.01) [46]. On the basis of the results of eight studies, fewer patients required vasopressors in the GDFT group [47,48,49,52,53,56,58], but statistical significance was not reached (RR: 0.90, 95% CI: 0.71 to 1.14). No influential study was detected by the leave-one-out sensitivity analysis (Appendix A). Cho et al. also provided data on intraoperative vasopressor requirement with no significant difference between the two GDFT groups as compared to the controls (44% and 24% versus 28%; *p* = 0.38) [46]. No significant difference was found in the intraoperative urinary output standardised for length of surgery (SMD: 5.69 mL h^−1^, 95% CI: −2.16 to 13.54 mL h^−1^). The leave-one-out sensitivity analysis did not identify any influential study (Appendix A). Serum lactate levels at the end of operation in the GDFT group were significantly lower compared to the N-GDFT group (WMD: −0.25 mmol L^−1^, 95% CI: −0.36 to −0.14). There was no evidence of heterogeneity (I2 = 42.7%, *p* = 0.175). Leave-one-out sensitivity analysis could not be performed due to the low number of studies.

### 3.4. Risk of Bias in Studies and Certainty of Evidence

Appendix A summarise the risk of bias assessment for all outcomes. All studies were judged as low risk or with some concerns.

A certainty of evidence table, including reasons for downgrading of the evidence level, is detailed in Appendix A. Certainty of evidence was considered very low for intraoperative fluid requirement, intraoperative vasopressor requirement, urinary output, time to first flatus, and length of hospital stay, whereas it was low for serum lactate levels at the end of the operation and time to first stool after the operation (Appendix A).

## 4. Discussion

The main findings of our meta-analysis are that patients treated with GDFT received less fluid during surgery, had lower serum lactate levels, both the first flatus and stool appeared earlier, and their hospital stay was also reduced compared to the N-GDFT-treated patients.

### 4.1. Summary of Evidence

Laparoscopic surgery may inflict profound effects on macro haemodynamic variables, resulting in elevated central venous and right atrial pressure, decreased cardiac output and stroke volume, and higher mean arterial pressure and systemic vascular resistance, due to elevated intraabdominal pressure and hypercarbia [4,5,59]. These can lead to decreased renal and splanchnic circulation, which are often responsible for unfavourable postoperative outcomes [5,60].

Adequate fluid management during surgery is of utmost importance to maintain adequate perfusion and oxygen delivery to the tissues. As both hypo- and hypervolaemia can be harmful, targeting fluid therapy to the patients’ individual needs is mandatory [61,62]. Fluid restriction per se, recommended in several guidelines as superior to liberal strategy [29,30], may reduce blood flow to the gastrointestinal tract, which may prolong the gastrointestinal recovery [5,63,64,65], impairing renal perfusion and leading to higher incidence of acute kidney injury after surgery [66]. Both effects can be precipitated during the pneumoperitoneum. Hypervolaemia and excessive fluid administration can also be harmful by causing interstitial oedema, which also impairs perfusion and oxygen uptake [67], which may lead to higher chance to surgical postoperative morbidity [68].

Conventional variables, such as heart rate and blood pressure, cannot predict fluid responsiveness and tell us little about tissue perfusion. Advanced haemodynamic monitoring (invasive, less invasive, non-invasive) has been tried and tested intensively for decades [69], but discussing these are beyond the scope of the current article. One advantage of using haemodynamic monitoring is being able to implement GDFT [62,70]. In the current meta-analysis, we included studies that compared GDFT to N-GDFT in patients undergoing laparoscopic surgery.

Our results suggest that GDFT may lead to a shorter length of hospital stay. This finding was significant and can also be considered compelling in the clinical practice. This observation is in accordance with previously reported results [71,72,73]. However, no significant difference was detected in those studies that implemented the ERAS. Further investigations are necessary whether GDFT combined with ERAS or other fast-track surgery protocols provides additional benefit of shorter hospitalisation or not.

One of the most important results of the current meta-analysis is that GDFT was associated with faster gastrointestinal recovery as indicated by shorter time to first stool. Although this outcome may be seen as of particular importance only after bowel surgery, there is substantial evidence to support that any abdominal surgery that applies pneumoperitoneum can lead to impaired bowel function [5,74]. Former studies suggested that the best way to evaluate the functional recovery of the gastrointestinal tract after surgery is the time to tolerate solid food and to pass the first stool. This is in alignment with our findings. These findings suggest that using GDFT may help to individualise fluid management and had not been shown in the two previous meta-analyses [23,24].

Conventional monitoring of heart rate and blood pressure have been shown to be inadequate measures of perfusion in general, and hence normal values do not exclude splanchnic hypoperfusion causing decreased oxygen delivery, resulting in anaerobic glycolysis and accumulation of lactate. The latter is an important marker to detect insufficient oxygen supplementation [75]. Although the lactate levels at the end of the operation were in the acceptable therapeutic range in both groups, levels were lower in the GDFT patients as compared to the N-GDFT group, indicating that the lesser amount of intraoperative fluid administration did not cause underfilling and/or consequential hypoperfusion. Unfortunately, previous meta-analyses did not report on serum lactate levels directly following the operation. However, our findings are in accordance with that of Forget et al. [76]. In their study, significantly lower lactate levels were reported in the GDFT group (GDFT: 1.2 mmol L^−1^; CI: 1–1.4 CI versus N-GDFT 1.6 mmol L^−1^; CI: 1.2–2.0). It is important to note that they confined their investigation to the intraoperative period.

### 4.2. Strengths and Limitations

This is the first meta-analysis that has investigated the effects of GDFT versus N-GDFT specifically in laparoscopic abdominal surgery. Our study reflects on both physiological issues at the end of the operation and measures of gastrointestinal recovery, length of hospital stay, and overall complications and readmission rate. Furthermore, the trials originate from several countries and continents, which increases the representative value of the results. Finally, the studies included in our analysis were published mainly in the last five years, and hence our results provide data that have not been considered yet in recent guidelines.

Our meta-analysis also has some limitations. First, most of the trials were single-centre RCTs with a low number of patients, which probably decreases the external validity of the studies. This may explain the high heterogeneity of several analyses. Second, the applied haemodynamic monitoring technologies in the GDFT group and the fluid administration regimens in the controls showed great variability, which may also point out the high heterogeneity for length of hospital stay, time to first flatus, and intraoperative fluid requirement. Third, we were unable to perform subgroup analysis on the type fluids used (i.e., crystalloids vs. colloids) due to the quality and quantity of the data reported on the outcomes we investigated. Furthermore, we were unable to perform a quantitative analysis for the overall complication rates due heterogeneous reporting on complications.

## 5. Conclusions

To our knowledge, this is the first and most comprehensive meta-analysis to date that reports on the effects of intraoperative GDFT resulting in less intravenous fluid administration, lower postoperative lactate levels, and enhanced recovery of the gastrointestinal function, which may lead to reduced hospital stay in patients undergoing elective abdominal laparoscopic surgery. Whether GDFT would result in overall advantageous outcomes including healthcare costs as compared to the generalised “fluid restriction” strategy recommended by the ERAS protocols in laparoscopic surgery has to be determined by further research.

## Figures and Tables

**Figure 1 jpm-12-00734-f001:**
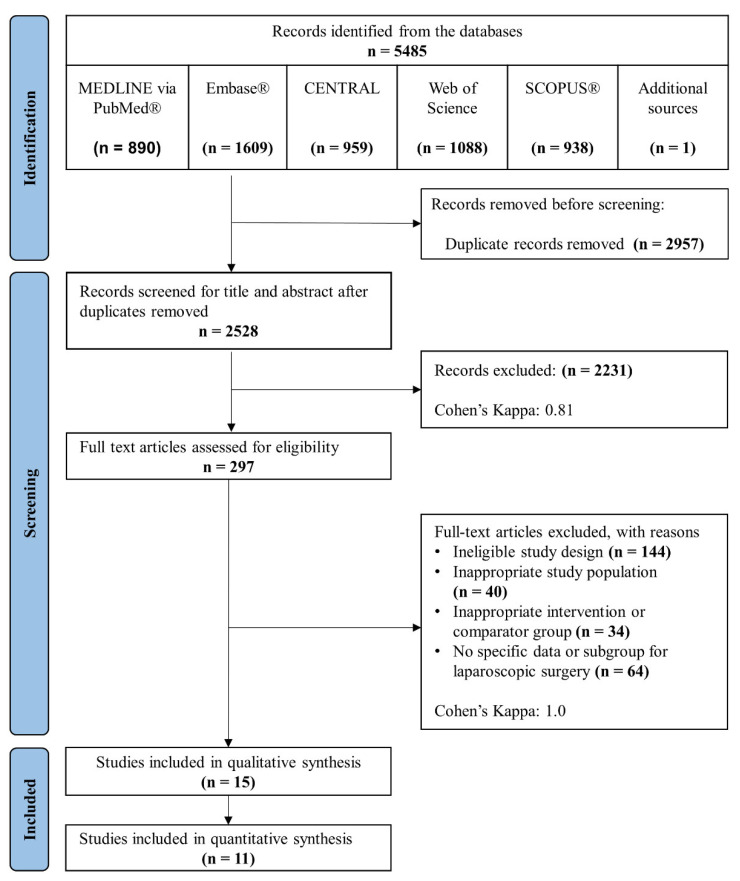
PRISMA flowchart of selection.

**Figure 2 jpm-12-00734-f002:**
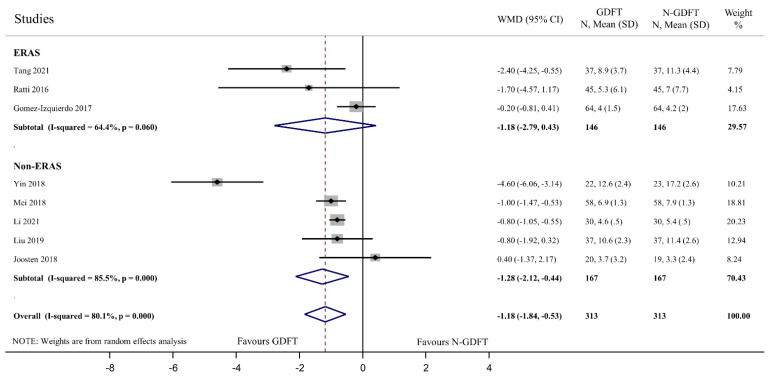
Length of hospital stay (days). Length of hospital stay was significantly shorter in patients who received GDFT (WMD = −1.18 days; 95% CI = −1.84 days to −0.53 days) and also in the non-ERAS subgroup (WMD = −1.28 days; 95% CI = −2.12 days to −0.44 days). However, in the ERAS subgroup, our result was not significant (WMD = −1.18 days; 95% CI = −2.79 days to 0.43 days). Heterogeneity was high both in overall and in the non-ERAS group (I-squared = 80.1%; *p* < 0.01 and I-squared = 85.5%; *p* < 0.01), and moderate in the ERAS subgroup (I-squared = 64.4%; *p* = 0.06).

**Figure 3 jpm-12-00734-f003:**
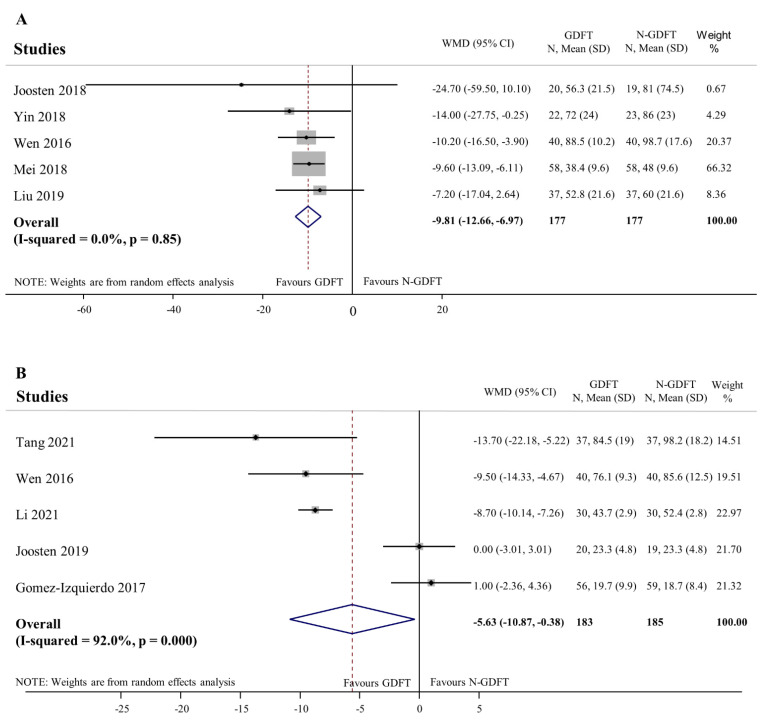
Time to first stool and time to first flatus. Time to first stool (**A**) was significantly reduced in patients receiving GDFT compared to the controls (WMD = −9.81 h; 95%; CI = −12.66 h to −6.97 h). No evidence was found for heterogeneity (I-squared = 0.0%; *p* = 0.85). Time to first flatus (**B**) was significantly shortened in the GDFT group compared to the controls (WMD = −5.63 h; 95% CI = −10.87 h to 0.38 h). High heterogeneity was detected (I-squared = 92.0%; *p* < 0.01). WMD: weighted mean difference, SD: standard deviation, GDFT: goal-directed fluid therapy, N-GDFT: non-goal-directed fluid therapy, CI: confidence interval. *p* < 0.1 was considered significant.

**Figure 4 jpm-12-00734-f004:**
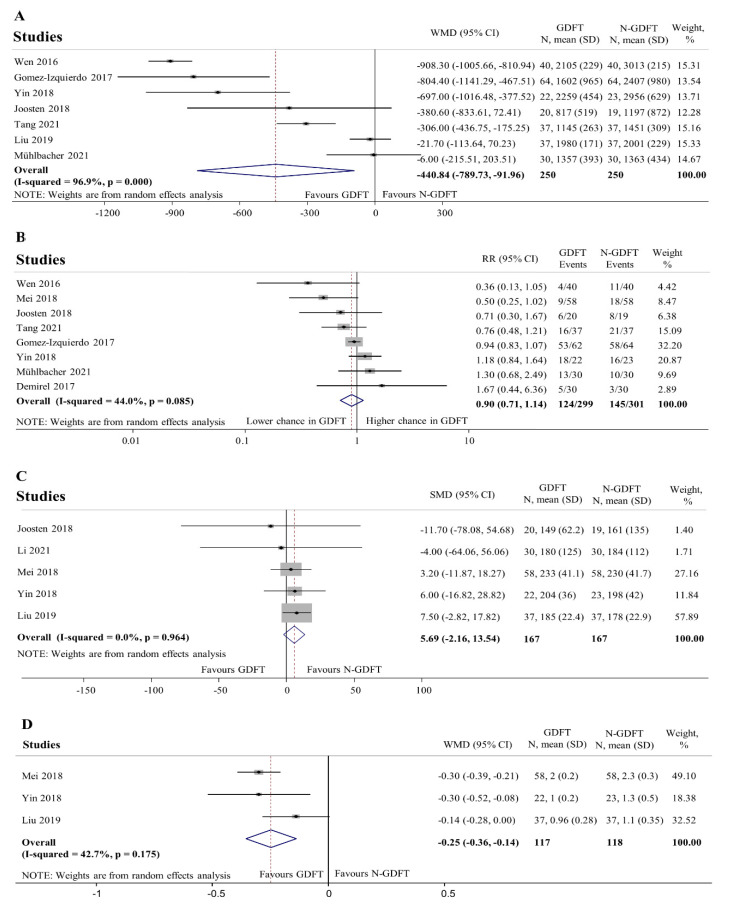
Clinical outcomes at the end of operation. Intraoperative fluid requirement (**A**) was significantly lower (WMD = −440.84 mL; 95% CI: −789.73 mL to −91.96 mL) in the GDFT group. High heterogeneity was detected (I-squared = 96.9%, *p* < 0.01). There was no significant difference in the number of patients requiring vasopressors intraoperatively (**B**) between the goal- and the non-goal-directed groups. (RR = 0.90; 95% CI = 0.71 to 1.14). Low heterogeneity was found (I-squared = 44.0%; *p* < 0.01). There was no significant difference in intraoperative urinary output standardised for length of surgery (**C**) between the two groups (SMD = 5.69 mL h^−1^; 95% CI = −2.16 mL h^−1^ to 13.54 mL h^−1^). Data were not considered heterogeneous (I-squared = 0.0%; *p* = 0.96). Serum lactate levels (**D**) were significantly lower in the GDFT group compared to N-GDFT (WMD = −0.25 mmol L^−1^; 95% CI −0.36 mmol/ to −0.14 mmol L^−1^). There is no evidence for heterogeneity (I-squared = 42.7%; *p* = 0.18). WMD: weighted mean difference, SMD: standardised mean difference, RR: risk ratio, SD: standard deviation. GDFT: goal-directed fluid therapy, N-GDFT: non-goal-directed fluid therapy, CI: confidence interval. *p* < 0.1 was considered significant.

**Table 1 jpm-12-00734-t001:** Characteristics of included studies.

Author (Year)	Type of Surgery	Preoperative Fluid Protocol	Intraoperative Fluid Protocol						Postoperative Fluid Protocol	Primary Outcome
			Haemodyamic Technology	Primary Goal	Bolus	Type of Fluid	Basis	Type of Fluid		
Brandstrup (2012) [44]	Elective laparoscopic colorectal resection	0.9% saline UD *	Oesophageal Doppler	SV < 10%	200 mL	VOLUVEN^®^	Replacement of lost blood volume only	VOLUVEN^®^	daily 2000 mL	Overall postoperative complications
			N-GDFT		200 mL	VOLUVEN^®^	Replacement of lost blood volume only	VOLUVEN^®^		
Calvo-Vecino (2018) [45]	Laparoscopic gastrointestinal, urological, gynaecological	N/A	Oesophageal Doppler	SV < 10%	250 mL	VOLUVEN^®^, Lactated Ringer	0 mL kg^−1^ bw^−1^	None	N/A	Moderate or severe postoperative complications
			N-GDFT		AAE	VOLUVEN^®^, Lactated Ringer	3–5 mL kg^−1^ bw^−1^	Lactated Ringer		
Cho (2021) [46]	Laparoscopic sleeve gastrectomy	4/2/1	Arterial waveform-derived	SVV < 10%	100 mL	6% hydroxyethyl starch 130/0.4	4 mL kg^−1^ bw^−1^	Lactated Ringer or Saline 0.9%	N/A	Postoperative nausea and vomiting
			Arterial waveform-derived	SVV < 10%	100 mL	Lactated Ringer	4 mL kg^−1^ bw^−1^	Lactated Ringer or Saline 0.9%		
			N-GDFT		AAE	6% hydroxyethyl starch 130/0.4	4 mL kg^−1^ bw^−1^	Lactated Ringer or Saline 0.9%		
Demirel (2017) [47]	Laparoscopic RYGB surgery	N/A	Pulse oximetry	PVI < 14%	250 mL	Gelofusine^®^	2 mL kg^−1^ bw^−1^	0.9% NaCl or Lactated Ringer	N/A	Perioperative lactate, creatinine levels, hemodynamic variables
			N-GDFT		250 mL	Gelofusine^®^	4–8 mL kg^−1^ bw^−1^	0.9% NaCl or Lactated Ringer		
Gomez-Izquierdo (2017) [48]	Laparoscopic colorectal	4/2/1	Oesophageal Doppler	SV < 10%	200 mL	VOLUVEN^®^	1.5 mL kg^−1^ bw^−1^	Lactated Ringer	1.5 mL kg^−1^/bw^−1^/h^−1^ In PACU 15 mL h^−1^ in Surgical Department	Primary postoperative ileus
			N-GDFT		5 mL kg^−1^ bw^−1^	VOLUVEN^®^	4/2/1 Rule	Lactated Ringer		
Joosten (2018) [49]	Laparoscopic colorectal, gynaecological, urological	N/A	Arterial waveform-derived	SVV < 13%	100 mL	PlasmaLyte^®^	0 mL kg^−1^ bw^−1^	None	N/A	Percentage of intraoperative time spent within defined haemodynamic targets (CI ≥2.5 L/min/m^2^ and/or an SVV <13%)
			N-GDFT		AAE	6% hydroxyethyl starch 130/0.4	4 mL kg^−1^ bw^−1^	PlasmaLyte®		
Li (2021) [50]	Laparoscopic radical resection of lower cervical cancer	N/A	Arterial waveform-derived	SVV < 13%	250 mL	6% hydroxyethyl starch 130/0.4	500 mL	Lactated Ringer	N/A	Appearance of first bowel sounds, time to first flatus, lengths of hospital stay, incidence of postoperative nausea and vomiting
			N-GDFT		AAE	6% hydroxyethyl starch 130/0.4	N/A	Lactated Ringer		
Liu (2019) [51]	Laparoscopic colorectal	5 mL kg^−1^ bw^−1^ before anaesthesia	Arterial waveform-derived	SVV < 13%	200 mL	Colloid solution UD	2 mL kg^−1^ bw^−1^	Lactated Ringer	N/A	Haemodynamic variables and tissue oxygen saturations intraoperatively and at the end of operation
			N-GDFT		AAE	Colloid solution UD	5 mL kg^−1^ bw^−1^	Lactated Ringer		
Mei (2018) [52]	Laparoscopic precision hepatectomy	N/A	Arterial waveform-derived	SVV < 13%	3 mL kg^−1^ bw^−1^	Colloid solution UD	6–10 mL kg^−1^ bw^−1^	Crystalloid UD	N/A	MAP, SVV, CVP, and lactate levels through the intraoperative period and at the end of surgery
			N-GDFT		10 mL kg^−1^ bw^−1^	Crystalloid UD	6–10 mL kg^−1^ bw^−1^	Crystalloid UD		
Mühlbacher (2021) [53]	Laparoscopic gastric bypass	500 mL Lactated-Ringer	Oesophageal Doppler	SV < 10%	250 mL	Lactated Ringer	2 mL kg^−1^ bw^−1^	Lactated Ringer	AAE in PACU	Perioperative subcutaneous tissue oxygen tension (upper arm)
			N-GDFT		AAE	Lactated Ringer	N/A	Lactated Ringer		
Ratti (2016) [54]	Laparoscopic liver resection	ERAS **	Arterial waveform-derived	SVV < 12%	N/A	Crystalloid UD	N/A	Crystalloid UD	ERAS **	Rate and reasons of conversion
			N-GDFT		N/A	Crystalloid UD	N/A	Crystalloid UD		
Senagore (2009) [55]	Laparoscopic colorectal	N/A	Oesophageal Doppler	SV < 10%	300 mL	Lactated Ringer	5 mL kg^−1^ bw^−1^	Lactated Ringer	N/A	Length of hospital stay
			N-GDFT		AAE	6% hydroxyethyl starch 130/0.4/, Lactated Ringer	5 mL kg^−1^ bw^−1^	Lactated Ringer		
Tang (2021) [56]	Laparoscopic radical gastectomy	250 mL warm sugar water per os	Arterial waveform-derived	SVV < 13%	250 mL	6% hydroxyethyl starch 130/0.4	N/A	Crystalloid UD	N/A	Incidence of postoperative complications
			N-GDFT		AAE	6% hydroxyethyl starch 130/0.4	N/A	Crystalloid UD		
Wen (2016) [57]	Laparoscopic gastrectomy	N/A	Arterial waveform-derived	SVV < 13%	3 mL kg^−1^ bw^−1^	6% hydroxyethyl starch 130/0.4	5 mL kg^−1^ bw^−1^	Lactated Ringer	N/A	Changes of haemodynamic variables and application of vasoactive drugs
			N-GDFT		5 mL kg^−1^ bw^−1^	6% hydroxyethyl starch 130/0.4	7 mL kg^−1^ bw^−1^	Lactated Ringer		
Yin (2018) [58]	Laparoscopic colorectal	N/A	Bioreactance	SVV < 13%	250 mL	6% hydroxyethyl starch 130/0.4	8 mL kg^−1^ bw^−1^	Saline UD	N/A	Moderate or severe postoperative complications within 30 days
			N-GDFT		250 mL	6% hydroxyethyl starch 130/0.4	8 mL kg^−1^ bw^−1^	Saline UD		

Included in systematic review only. Included both in the quantitative and qualitative synthesis. *: if fluid intake was under 500 mL; **: according to ERAS protocol for liver surgery; 4/2/1: 4 mL per kilograms of bodyweight for the first 10 kg, 2 mL kg^−1^ bw^−1^ to the second 10 kg, 1 mL^−1^ kg^−1^ bw^−1^ to the other kg bw^−1^. Abbreviations: N-GDFT: non-goal-directed fluid therapy, SV: stroke volume, SVV: stroke volume variation, PVI: Pleth Variability Index, UD: undetermined, AAE: according to the anaesthetist evaluation, N/A: data not available, PACU: post-anaesthesia care unit, CI: Cardiac Index, CVP: central venous pressure, NaCl: natrium chloride, RYBG: Roux-en-Y gastric bypass surgery, MAP: mean arterial pressure, ERAS: enhanced recovery after surgery, bw: bodyweight.

## Data Availability

The datasets used and/or analysed during the current study available from the corresponding author on reasonable request.

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
