# Peer review of "Goal-Directed Fluid Therapy Enhances Gastrointestinal Recovery after Laparoscopic Surgery: A Systematic Review and Meta-Analysis"

_jpm, 2022, doi:10.3390/jpm12050734_

Round 1

Reviewer 1 Report

This study is a meta-analysis of the effect of GDFT on N-GDFT in laparoscopic abdominal surgery.

The presence of a lot of heterogeneity is a big limitation of this study, but the authors are aware of this point and it is mentioned in the Discussion. We look forward to future research.

I did not understand the following points in the interpretation of the results and would like the interpretation to be carefully added.

1)  Although Figures S17-S22 are available, they are not mentioned in the text. The authors do not refer to the results in Supplementary Figure and repeat in a footnote that "There is no study the omission of which would change the statistical significance." They simply repeat in a figure legend. This is dishonest and they should properly cite Figures S17-S22 in their interpretation of Figures 2-4 so that the text shows how they made their decision based on the results of the sensitivity analysis.

Reviewer 2 Report

Dear Editor, 

  This is an interesting systemic review regarding target fluid therapy after laparoscopic surgery. The followings are my comments.

  1. Figure 1,  5485 articles includes and 2832 records removed, but the number of screened article is 2528. It didn't match, please check
  2. Table 1, different types of surgery were  included for analysis. One of the end point, such as first stool pass may not be appliable to non-bowel surgery. Have the authors tried to spit the type of surgery to evaluate the outcome ?
  3. Table 1, differnet types of fluids were used. Have the authors tried to evaluate the effect of colloid vs crystalloid fluid on the study outcome ?
